# Sedentary Behaviors of a School Population in Brazil and Related Factors

**DOI:** 10.3390/ijerph17196966

**Published:** 2020-09-23

**Authors:** José Antonio Ponce-Blandón, María Eduarda Deitos-Vasquez, Rocío Romero-Castillo, Diogo da Rosa-Viana, José Miguel Robles-Romero, Jussara Mendes-Lipinski

**Affiliations:** 1Centro Universitario de Enfermería de Cruz Roja, Universidad de Sevilla, 41009 Sevilla, Spain; Japonce@cruzroja.es (J.A.P.-B.); jmrobles@cruzroja.es (J.M.R.-R.); 2Departamento de Enfermagem, Universidade Federal do Pampa, Uruguaiana 1650, Bagé, Brazil; Maria.eduardadeitos@gmail.com (M.E.D.-V.); jussaralipinski@gmail.com (J.M.-L.); 3Departamento de Enfermagem, Universidade Federal Rio Grande do Sul (UFRGS), Porto Alegre 90040-000, Brazil; Diogoviana95@yahoo.com.br

**Keywords:** eating habits, healthy lifestyle, sedentary behavior, exercise, child, child behavior, schools

## Abstract

*Background*. Overweight and obesity arise from a complex range of genetic, environmental, behavioral, educational, and socioeconomic factors. The present study explored the sedentary practices and some life habits related to health among children from the school population of Uruguaiana (Rio Grande do Sul, Brazil). *Methods:* A cross-sectional descriptive study was conducted to characterize the life habits of the school boys and girls from fourth grade (9–10 years old). *Results:* A total of 470 fourth-grade boys and girls (9–10 years old) participated in the study. As regards the variables linked to habits, 24% of the boys and girls answered they had not had breakfast the day they completed the questionnaire and 51.8% stated they did not have breakfast any given day of the week. Regarding sedentary habits, 25.3% of children watched TV or played video games five or more hours a day and 9% rarely played sports with their parents or caregivers. Statistical significance was recorded between “number of hours watching TV and playing video games” and “playing sports with parents or caregivers” (*p* < 0.05). *Conclusions:* Association between the times spent watching TV or playing video games and the practice of physical exercise in the family proves once again the importance of the family in education for the health of children. The school provides direct access to schoolchildren and their parents to launch numerous health education programs.

## 1. Introduction

According to the World Health Organization, childhood obesity is a public health problem, whose prevalence has increased in recent years with more than 100 million affected in 2015 [1,2]. If current trends continue, in 2022 there will be more children with obesity than with underweight [3]. Overweight and obesity in children cause significant impacts on the physical and mental health of children, with a tendency toward obesity in adulthood and suffering from noncommunicable diseases such as diabetes and cardiovascular diseases [4,5]. In England, a third (33.5%) of children aged 10 years are classified as overweight or obese [6]. In Latin America, it is estimated that between 42.4 and 51.8 million children and adolescents are overweight or obese, representing 20 to 25% of the total population of children and adolescents [7]. Specifically, in Brazil, one-third of children in the population between 5 to 9 years old are overweight [8]. 

Overweight and obesity arise from a complex range of genetic, environmental, behavioral, educational, and socioeconomic factors. Poor eating habits and poor physical activity increase the probability that children will become obese adolescents [9]. The sedentary lifestyle, encouraged by the rapid evolution of technology, has been accentuated in recent years [10]. Life habits have been modified, especially in children who have adopted behaviors that predispose to minimal energy expenditure, close to a quiescent condition. Children spend many hours in front of television, playing video games, or managing social networks from their mobile phones [11]. According to a survey carried out in Brazil, entitled National Student Health Survey, the prevalence of adolescents exposed to at least two hours a day of television is 78% [12].

Individual or intrapersonal factors that may influence physical activity relate to individual characteristics and choices. This includes knowledge, skills, self-efficacy, and individual socioeconomic circumstance. Children living in residential homes have reported lower scores in perception of physical activity ability/success. Lack of money is a major barrier to involvement in leisure and social activities, with the most frequent activity being sport [13]. Very little is known about how various interpersonal processes and primary social groups (peers, social workers, family, and foster caregivers) that provide support and role modeling influence activity choices on participation for children. The presence of friends has a positive influence on the desire to engage in a particular activity more often [14]. Two studies have reported on institutional factors. These include the rules, regulations, policies, and structures of institutions like schools that constrain or promote certain behaviors such as physical activity [15].

There is scientific evidence highlighting the association between the number of hours that television is viewed and the omission of important meals of the day, such as breakfast, and also with the consumption of hypercaloric foods, high in sugars, carbohydrates, and saturated fats [5,16]. The omission of breakfast has also been linked to a higher body mass index and an increase in leisure time in sedentary activities [17]. One of the strategies to control obesity in Brazil is through the National Food and Nutrition Policy (PNAN), updated by Ordinance No. 2,715 /2011, which aims to improve the diet, nutrition, and health conditions of the Brazilian population, through actions in order to promote adequate and healthy eating practices, food, and nutritional surveillance, especially the prevention of health problems related to an inadequate diet [18].

Strategies to accelerate obesity prevention include modifying the child’s environment for healthy eating and physical activity [19]. For this, it is essential to analyze the characteristics and life habits of the children’s population. This study tried to know the sedentary practices and some life habits related to health among children from the school population of Uruguaiana (Rio Grande do Sul, Brazil), identifying the sociodemographic factors that can influence it. The main hypothesis of the study was that the exercise with the family reduces the hours of exposure to television and video games in children.

## 2. Materials and Methods 

### 2.1. Study Design and Sample

A cross-sectional descriptive study was conducted to characterize the life habits of the school boys and girls from fourth grade (9–10 years old) in the city of Uruguaiana (Rio Grande do Sul, Brazil). Uruguaiana is a city of about 129,000 inhabitants, located at the western end of Rio Grande dos Sul. According to data from Brazilian Institute of Geography and Statistics, pediatric outpatient care is performed in hospitals and outpatient care is exclusively on demand. At present, no child health promotion services are offered, or child health monitoring services once the expectant mothers are discharged from the hospital after giving birth.

In order to obtain the sample, a random sampling was performed by clusters of the 24 urban public schools of the city. To calculate the minimum sample size, it was taken into account the number of schools, the number of children enrolled in the fourth grade, and the number of classes. There was a total of 810 children, 39 classes, and 26 schools. For a confidence level of 95% and an error range of 5%, the minimum size would be 264 children. 

### 2.2. Data Collection and Statistical Analyses

A transcultural adaptation was conducted of the abridged questionnaire of Eating Habits of the School Population of the “PERSEO” program (Pilot School Reference Program for Health and Exercise against Obesity) by the Spanish Society of Community Nutrition, validated in the Spanish population and translated into Portuguese [20]. A draft was sent to three social scientists and experts in this area who were external to the research team. They performed a review of the translated questionnaire, considering aspects such as item comprehension (both questions and possible answers) in relation to the objectives of the research. The items related to sedentary habits were selected. Questions included single, multiple, and open answers that were grouped into different variables:Sociodemographic variables: Age, gender, family situation, and the parents’ working situation and their types of occupation, according to the Brazilian classification of occupations [21].Anthropometric variables: Weight, height, and body mass index.Variables of sedentary habits: Screen time during lunch or dinner and sports activities with parents.

The questionnaire was applied to the selected sample in 12 schools. The headships of all the selected school centers were contacted. The data collection day and time were scheduled, and the boys’ and girls’ parents were asked to sign the informed consent. Two educated researchers, accompanied by the teacher of the group, explained the instructions to complete the questionnaire to the children and clarified any doubt. To record and analyze the data, a questionnaire was written in the specific tool from the EpiInfo application, Version 7.2.1.0. (Epi, Atlanta, GA, USA) This tool was also employed to perform the statistical analysis.

The frequency distributions of the main qualitative variables were calculated in the descriptive analysis, as well as the measurements of central tendency and dispersion of the quantitative variables. For the hypothesis contrast, the chi-square test or Fisher’s exact test and student’s t test were used, as it suited the case. The accepted statistical significance level for the hypothesis contrast was set at *p* < 0.05.

The researchers obtained documentary authorization from the Regional Coordinating Office for Education of Rio Grande do Sul and from the Research Ethics Committee of the University of Unipampa (Code: 05.2017). During the whole process of data collection, children’s anonymity was guaranteed and the ethical principles for medical research in human beings described in the latest revision of the Declaration of Helsinki were applied [22].

## 3. Results

A total of 470 fourth grade boys and girls (9–10 years old) participated in the study. Of the total participants, 46% were boys and 54% were girls and the mean age of the participating children was 9.8 years old (SD = 0.74). Thirty-eight percent were children from the morning group, 56.7% were from the afternoon group, and 5.3% were from the integrative group (morning and afternoon shifts). Of the children, 99.6% were Brazilian and 64.2% of the children stated they lived with their parents all the time, whereas the rest had different family situations.

The mean number of brothers/sisters reported by the children was 2.07, 83.5% of the children stated that their father worked, and 63.7% that their mother also worked. The main occupation of 51.4% of the mothers who worked was in the area of services and in retail sales in stores or supermarkets. Of the fathers, 77.1% shared this same occupation. The main sociodemographic and characteristics’ descriptive results of the sample are summarized in Table 1. 

As regards the variables linked to habits, referred to in Table 2, 24% of the boys and girls answered they had not had breakfast the day they completed the questionnaire. Of those who had breakfast, 34.9% had taken milk or eaten legumes, dried fruits, or eggs; 24.4% fruits or natural fruit juices; 10.4% bread, rice, or potatoes; 6.9% cold meats and processed foods; and 3.1% candies, pastries, or salted snacks. With respect to the frequency of weekly breakfast, 51.8% stated they did not have breakfast any given day of the week, 49.3% had breakfast every day with their mother or father, and 14.2% never had breakfast with their mother/father. Regarding sedentary habits, 25.3% of children watched TV or played video games five or more hours a day, 55.3% watched TV at lunch and dinner, and 9.1% rarely played sports with their parents or caregivers.

The results related to the hypothesis contrast, obtained with the chi-square test, are presented in Table 3. The variable “screen time (hours)” became a dichotomous variable: Less than 2 h and 2 or more hours’ screen time. In comparison with other variables (sex, family situation, parents’ work, and breakfast habits), only statistical significance was recorded with two variables. There was a significant difference between screen time and sex (*p* < 0.05) and playing sports with parents or caregivers (*p* < 0.05). A multivariable regression model was made in order to identify the independent associations of screen time with physical activities controlling for age, sex, and sociodemographic variables. The result was not statistically significant (Table 4).

## 4. Discussion

In the present study, a descriptive analysis of the data was carried out and various sociodemographic hypotheses were tested, as well as dietary habits and sedentary habits compared to the number of hours they watched television and/or played video games. The National Student Health Survey, carried out in Brazil, showed that 78% of adolescents were exposed to at least two hours a day of television [12]. In our study, the population was school-aged between 9 and 10 years and the percentage of boys and girls who watched television or played video games at least 2 h a day was 55%.

Stahlmann et al. observed that children from single-parent families reported longer hours of screen time (*p* < 0.001) [23]. In our study, no significant difference was observed in this regard (*p* = 0.476). However, the highest percentage of boys and girls who had screen time over two hours was represented by those who lived with their father and a new partner (66.6%), followed by those who lived alone with their mother (46.7%).

There is scientific evidence linking the amount of screen time with the omission of important meals during the day such as breakfast [16]. Specifically, in our study, no significant differences were obtained between screen time and the weekly frequency of breakfast. This may be due to the fact that the problem in this population is more of education, habits, and family education than the influence of television. However, it was found that almost half of children who do not have breakfast had screen time of over two hours a day.

In this study, only 18% of boys and girls had breakfast every day of the week, a low rate compared to other studies carried out in Argentina [24], Spain [25], and Canada [26], where 75%, 77.5%, and 85.5%, respectively, had breakfast daily. However, in the study carried out in Spain [25], 5.2% had fresh juices or whole fruits for breakfast; on the contrary in our study, this breakfast (fresh juices or whole fruits) was taken by 24.4%. In the study carried out in Argentina, 16% had a low-quality breakfast, based on industrial pastries and sugary foodstuffs, while in Italy [27], 31.3% of children had this type of breakfast. By comparison, among Uruguaiana schoolchildren, only 3.1% had industrial pastries or sugary foodstuffs for breakfast. This was particularly noteworthy, so we can conclude that while there were fewer children who used to have the habit of a daily breakfast, those who did ate a healthier breakfast, mainly based on milk, bread, eggs, and fruit or fresh fruit juices. The problem in this case was the lack of the breakfast habit, which may be related to educational and socioeconomic factors due to scarcity of resources.

In previous studies, screen time was associated with a higher intake of “fast food”, foodstuffs and foods rich in saturated fat, refined sugar, food colors, and preservatives [28,29,30]. Overexposure to television and video games, in addition to leading to a sedentary lifestyle, exposes children to the advertising of unhealthy products [31,32]. In this study, no statistical significance was obtained between screen time and the type of breakfast that children had. This may be due to the fact that the advertising of processed products and fast food in Uruguaiana is less than in other cities. Some of the most recognized and publicized products by international celebrities are expensive and difficult to access for a population with limited resources.

Statistical significance was obtained between screen time and the frequency of practicing sports activities or physical exercise with their parents or other caregivers. This finding is similar to other ones found in previous studies, in which the important role of the family appears to be related to sedentary behaviors of children [33,34,35]. In fact, parents’ lifestyle habits are the main predictors of screen time by schoolchildren [36,37,38]. Bjelland et al. concluded that parental support and also their teaching and accompaniment in physical exercise decreased the screen time among European children [39,40,41,42]. A multivariate regression was performed between screen time and some sociodemographic variables and no statistical significance was obtained.

Another of the determining factors related to screen time is age and gender [43,44]. Regarding age, no differences were observed in this study, because there was hardly any variability, but on the other hand, there was a significant association between the screen time and gender. Even though the proportion of girls was lower, boys were more likely to have screen time during the day than girls, as 58.12% of them had screen time of less than two hours a day. According to Saunders et al., boys reported more screen time (two or more hours) than girls and total sedentary behavior [45]. Physical competence, motivation, and confidence were negatively associated with all modes of sedentary behavior. Knowledge and understanding about the benefit of physical exercise were negatively associated with screen-based modes of sedentary behavior and positively associated with nonscreen, sedentary behavior [46]. Public health interventions should continue to target screen-based sedentary behaviors, given their potentially harmful associations with important aspects of physical literacy [45].

Surveys in Brazil have shown that smoking, obesity, and unhealthy diets are more frequent in individuals with low educational attainment. The Brazilian phone surveillance system shows that leisure-time physical activity is most frequent in young adults, men, those with high educational attainment, and people who live near public spaces with equipment for physical activity. The progress in physical activity reflects improvements in active transportation, health education, communication strategies, and physical activity interventions funded by the Brazilian Ministry of Health. The Brazilian government launched a strategic plan to reduce exposure to risk factors, such as smoking, consumption of alcohol, physical inactivity, and salt intake. The government took account of the results of evaluation studies of community interventions to promote physical activity classes in community settings [47].

Most of the scientific community agrees that the growth in worldwide prevalence of obesity during the last decades is due to profound changes in our traditional lifestyle. Today people are more sedentary. Furthermore, many countries are losing their traditional diets by others precooked, sugary, and rich in animal fats’ products [48]. Several studies have shown the relationship between an increase of several sedentary behaviors, for instance, screen time, and weight gain [49]. A novel risk factor in children and adolescents is TV availability in the bedroom [50]. Interventions aimed at reducing screen time have been a focus of childhood obesity prevention and treatment. However, the evaluation of their effectiveness needs to be taken into account in order to develop successful prevention programs [51].

Although the technological revolution has been of great benefit to many populations throughout the world, it has come at a major cost in contribution of physical inactivity to the worldwide epidemic of noncommunicable disease. Evidence from Brazil suggests that although prevalence of physical inactivity increased greatly in people with low income, no significant differences were reported in those with higher earnings. As public health efforts to increase physical activity and decrease sedentary time proceed, standardized physical activity surveillance procedures need to be implemented broadly and repeatedly. These measures are necessary to understand which intervention strategies work and to identify populations at greatest risk [52].

### Limitations

The intention of this study was to evaluate the sedentary behaviors of schoolchildren and their correlation with various sociodemographic and lifestyle factors. Children were asked to fill in the data related to their weight and height in order to determine their body mass index. There was a very low response rate because children did not know these data. The body mass index could have been compared to sedentary habits, if we had had a higher response rate. Therefore, as a proposal for improvement for the next study, it is recommended to personally size and weigh the children. It is also intended to include parents to expand the variables, as well as to contrast the information reported by the children.

## 5. Conclusions

In this study, an important finding was obtained, the association between the time spent watching television or playing video games and the practice of physical exercise in the family. Once again, the importance of the family in education for the health of children is highlighted. The school environment provides direct access to schoolchildren and their parents to launch numerous health education programs based on physical exercise and a healthy diet.

## Figures and Tables

**Table 1 ijerph-17-06966-t001:** Sociodemographic and characteristics of the sample.

Qualitative Variables	Categories	*n*	%	95% CI
Sex	Women	230	54.0	(49.2–58.6)
	Men	196	46.0	(41.3–50.7)
Class	Morning class	173	38.0	(33.7–42.5)
	Afternoon class	258	56.7	(52.1–61.2)
	Integral class	24	5.3	(3.6–7.7)
Nationality	Brasil	468	99.6	(98.4–99.8)
	Others (Argentina, Uruguay)	2	0.4	(0.04–1.2)
Family living situation	Alone with mother/father	96	21.9	(18.7–25.1)
	With mother/father and her/his new partner	29	67	(4.2–9.2)
	With father and mother	280	64.2	(59.6–68.5)
	With other adults	31	7.1	(5.0–9.9)
	Yes, he works	375	83.5	(79.8–86.6)
	He doesn’t work	30	6.7	(4.7–9.4)
	I don’t know/I haven’t contact with my father	44	9.8	(8.2–11.4)
Type of father’s occupation	0 Armed forces, Police, fire and military	1	0.2	(–0.21–0.6)
	1 Senior members of the government, managers of public interest organizations and companies and managers	1	0.2	(−0.21–0.6)
	2 Science and arts professionals	8	1.8	(1.5–2.3)
	3 Level technicians	11	2.4	(1–3.8)
	4 Administrative service workers	25	5.6	(3–7.6)
	5 Service workers, retail salespeople in stores and markets	199	44.2	(39.6–48.8)
	6 Agricultural, forestry, hunting and fishing workers	0	0.0	(0.0–0.0)
	7 Industrial and service production workers	6	1.3	(0.3–2.4)
	8 Maintenance and repair workers	3	0.7	(−0.1–1.4)
Mother work	Yes, she works	288	63.7	(57.9–66.9)
	She doesn’t work	144	31.8	(27.7–36.3)
	I don’t know/I haven’t contact with my mother	20	4.4	(2.8–6)
Type of mother’s occupation	0 Armed forces, Police, fire and military	10	2.2	(0.8–3.6)
	1 Senior members of the government, managers of public interest organizations and companies and managers	3	0.7	(0–1.4)
	2 Science and arts professionals	5	1.1	(0–2.1)
	3 Level technicians	12	2.7	(1–4.1)
	4 Administrative service workers	25	5.6	(3–7.7)
	5 Service workers, retail salespeople in stores and markets	175	38.9	(34.4–434)
	6 Agricultural, forestry, hunting and fishing workers	18	4.0	(2.2–5.8)
	7 Industrial and service production workers	41	9.1	(6.4–11.8)
	8 Maintenance and repair workers	49	10.9	(8.0–13.8)

**Table 2 ijerph-17-06966-t002:** Habits of the sample.

Variables	Categories	*n*	%	95% CI
Breakfast today in the morning	Yes	342	76.0	(72.5–79.5)
	No	108	24.0	(18.8–28.2)
Breakfast classification	Sweets, pastries and salty snacks	14	3.1	(1.50–4.7)
	Red, processed and could meats	31	6.9	(4.5–9.3)
	Milk, lean meats, legumes, nuts, eggs	157	34.9	(30.5–39.3)
	Vegetables and fruits	110	24.4	(20.5–28.4)
	Bread, pasta, rice, potatoes	47	10.4	(7.6–13.3)
Frequency of breakfast with mother or father	Everyday	222	49.3	(44.7–54.0)
	4–6 days per week	29	6.4	(4.2–8.7)
	1–3 days per week	66	14.7	(11.4–17.9)
	Less than once a week	50	11.1	(8.2–14.0)
	Never	64	14.2	(10.9–17.5)
Breakfast frequency	Everyday	81	18.0	(14.4–21.6)
	5–6 days per week	81	18.0	(14.4–21.6)
	1–4 days per week	31	6.9	(4.5–9.2)
	None	233	51.8	(47.1–56.4)
Screen time per day	None	47	10.0	(7.3–12.7)
	1 h/day	145	30.9	(26.6–35.1)
	2 h/day	56	11.9	(8.9–14.8)
	3 h/day	44	9.4	(6.7–12)
	4 h/day	39	8.3	(5.8–10.8)
	5 or more hours/day	119	25.3	(21.4–29.3)
Screen time at lunch or dinner per week	Everyday	260	55.3	(50.8–59.8)
	4–6 days/week	41	8.7	(6.1–11.3)
	1–3 days/week	64	13.6	(10.5–16.7)
	Less than 1 day/week	80	17.0	(13.6–20.4)
	Never	0	0.0	(0.0)
Sport frequency	Yes, most days	174	37.0	(32.6–41.4)
	Sometimes	139	29.6	(25.4–33.7)
	Only on weekends	34	7.2	(4.9–9.6)
	Rarely	43	9.1	(6.8–11.8)
	Never	0	0.0	(0.0)

**Table 3 ijerph-17-06966-t003:** Contingency table using the chi-squared test.

Variables	Screen Time (2 h/2 or More hours)	Statistical Value Pearson’s Chi-Squared	Level of Significance
Sex	Males	12.531	*p* = 0.028 *
	97 (50.3%)		
	96 (49.7%)		
	Women		
	132 (58.1%)		
	95 (41.9%)		
Family living situation	Alone with the mother	4.527	*p* = 0.476
	40 (53.3%)		
	35 (46.7%)		
	Alone with the father		
	14 (73.4%)		
	5 (26.3%)		
	With the mother and her new partner		
	11 (55%)		
	9 (45%)		
	With the father and his new partner		
	3 (33.3%)		
	6 (66.6%)		
	With father and mother		
	151 (54.5%)		
	126 (45.5%)		
	With other adults		
	17 (56.7%)		
	13 (43.3%)		
Father work	Yes, he works	1.964	*p* = 0.580
	198 (53.4%)		
	173 (46.6%)		
	He doesn’t work		
	17 (56.7%)		
	13 (43.3%)		
	I don’t know		
	13 (61.9%)		
	8 (38.1%)		
	I haven’t contact with my father		
	14 (66.6%)		
	7 (33.3%)		
Mother work	Yes, she works	3.977	*p* = 0.264
	163 (57.2%)		
	122 (42.8%)		
	She doesn’t work		
	70 (49%)		
	73 (51%)		
	I don’t know		
	7 (58.3%)		
	5 (41.7%)		
	I haven’t contact with my mother		
	4 (80%)		
	1 (20%)		
Breakfast frequency with parents	Everyday	7.131	*p* = 0.129
	124 (56.4%)		
	96 (43.6%)		
	4–6 days per week		
	22 (66.7%)		
	11 (33.3%)		
	1–3 days per week		
	38 (56.7%)		
	29 (43.3%)		
	Less than once a week		
	32 (61.5%)		
	20 (38.5%)		
	Never		
	27 (42.2%)		
	37 (57.8%)		
Breakfast frequency	Everyday	0.270	*p* = 0.966
	128 (53.8%)		
	110 (46.2%)		
	5–6 days/week		
	18(54.5%)		
	15 (45.5%)		
	1–4 days/week		
	46(56.8%)		
	35 (43.2%)		
	None		
	42(53.2%)		
	37 (46.8%)		
Breakfast classification	Sweets, pastries and salty snacks	0.78	*p* = 0.999
	8(57.1%)		
	6(42.9%)		
	Red, processed and could meats		
	17(54.8%)		
	14(45.2%)		
	Milk, lean meats, legumes, nuts, eggs		
	87(55.4%)		
	70(44.6%)		
	Vegetables and fruits		
	62(56.4%)		
	48(43.6%)		
	Bread, pasta, rice, potatoes		
	28(57.1%)		
	21(42.9%)		
Watch television at lunch or dinner	Everyday	1.118	*p* = 0.773
	147(56.5%)		
	113(43.5%)		
	4–6 days/week		
	23(56.1%)		
	18(43.9%)		
	1–3 days/week		
	31(49.2%)		
	32(50.8%)		
	Less than 1 day/week		
	44(55%)		
	36(45%)		
	Never		
	0(0%)		
Sports per day	Yes, most days	35.208	*p* = 0.019 *
	96(55.5%)		
	77(44.5%)		
	Sometimes		
	85(61.6%)		
	53 (38.4%)		
	Only on weekends		
	17(50%)		
	17(50%)		
	Rarely		
	16(37.2%)		
	27(62.8%)		
	Never		
	31(52.5%)		
	28(47.5%)		

* Indicates statistical significance.

**Table 4 ijerph-17-06966-t004:** Multivariate regression model.

Model	Sum of Squares	df	Mean Square	F	Statistical Significance
Regression	13,703	5	2741	0.822	0.535
Residual	1,283,969	385	3335		
Total	1,297,673	390			
	Beta	Sig.			
Sex	0.062	0.225			
Sport per day	0.047	0.351			
Mother’s work	0.003	0.059			
Father’s work	−0.076	0.145			

Dependent variable: Screen time per day. Predictors: Sport per day, sex, mother’s work, father’s work.

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
