# Peer review of "Sedentary Behaviors of a School Population in Brazil and Related Factors"

_ijerph, 2020, doi:10.3390/ijerph17196966_

Round 1

Reviewer 1 Report

  • General comments

This study is of interest as it addresses sedentary lifestyle patterns and correlates in school-age children (9-10 years of age). The objectives, the methods and the results are clear. The manuscript is well written. However, multivariable statistical models would be necessary in order to identify the independent associations of screen time with physical activities controlling for age, sex and socio-demographic variables. The discussion could be strengthened if multivariable statistics were available. Additionally, concrete recommendations for appropriate intervention strategies should be described. The format of the tables has to be modified.

  • Specific comments
  1. Sample size calculations: What variables were used? Statistical power may not be sufficient for some associations.
  2. Multivariable regression models would be necessary, as mentioned in the general comments.
  3. The authors will want to justify the choice of the study location, why only urban public schools were selected, and why only 9-10 y.o. students.
  4. Table 1 is not useful for an international readership.
  5. Tables 2-4 are much too long and not correctly formatted. They have to be shorter and be presented in a more conventional manner. Variable names can be simplified and some categories can be regrouped.
  6. Results with confidence intervals or standard deviations should not be repeated in the text when shown in tables. Only highlight the salient findings in the text.
  7. Instead of number of hours of television etc., we suggest simply using screen time (hours).
  8. Line 83: Translated INTO PORTUGUESE we assume.

Reviewer 2 Report

I received the manuscript entitled ‘Sedentary Behaviors of a School Population in Brazil and Related Factors’ for expertise. The authors explore the sedentary practices and some life habits related to health among children from the school population of Uruguaiana. The main outcome of the study was recorded between “number of hours watching TV and playing video games” and “playing sports with parents or caregivers” (p< 0.05). The study has a very interesting objective. However, in my opinion, the article needs some corrections in order to improve its quality.

Overall, the paper requires English correction to avoid small mistakes. For example, see remarks 4 & 5

Remark 1 (Introduction): It would be interesting for the authors to supplement the introduction with a short paragraph on barriers to active behavior in children and obese children.

Remark 2 (line 39): The authors note that the prevalence of obesity in children is increasing. I would like the authors to specify the prevalence rate in this case.

Remark 3 (lines 104-108): Authors must specify the number of the ethics committee's decision.

Remark 4 (table 1-first line): replace "participating ... simple" with "participating ... sample".

Remark 5 (table 2): In this table, the number of participants is 423. However, when the number of females and males is added together, the total is 426. In addition, in Table 1, there are 456 students participating in the sample. So I'd like to know specifically what the sample size was? In addition, authors should replace "Mens" with "Men". What is the meaning of “media” Do the authors mean median?

Remark 6 (Table 4): how were the comparisons made on the variable (Sports with mother, father, or other relatives)? Two by two? I don't understand at all. Indeed, since there are 5 subgroups, should an analysis of variance not be used? Finally, for reasons of homogeneity, the authors could replace "males" with "men".

Remark 7 (160-181): In this part of the discussion, the authors' results are inconsistent with those of the literature. I would like the authors to discuss further the possible explanations for the contradictory results in the literature.

Remark 8 (line 200): Instead of saying about 60%, I would like the authors to be more precise and make it clear that it is 58.12% of girls. However, I think the authors also need to discuss the results on the "2 or more hours" variable. Finally, I think that the authors must put their interpretation into perspective since there are 34 more girls. 

Remark 9: I think the discussion could be further enhanced if the authors consider the barriers to physical activity for children in interpreting the results for the variable “number of hours watching TV and playing video games. See DOI: 10.1123/jpah.2015-0410

Reviewer 3 Report

The article presents an important and relevant issue in the present scenario of Public Health. In general, it is a well-written article, with clear objectives, consistent results and adequate discussion. I suggest reviewing additional points like:

- Authors can better detail the study population and sampling methods.

- Remove rows from tables

- The Media and Standard Deviation can be presented throughout the text and removed from the table 2.

- Reduce and improve the presentation of Table 4

- I suggest to deepen the discussion of the importance of evaluating sedentary behaviors in addition to evaluating the practice of physical activity.

- The authors can better contextualize the reality of the study participants, in the light of social inequalities in Brazil.

- Authors should add to the discussion the increase in recent years of national / local public policies that favored access and the practice of free physical activity in public spaces. More details can be seen at https://doi.org/10.1016/S0140-6736(12)61041-1

- Recent evidence has pointed to the need for public health policies that encourage not only an increase in the level of physical activity, but also a reduction in Sedentary Behaviors, especially in young people.  Authors need to discuss this point better. More details can be seen at https://doi.org/10.1016/S0140-6736(12)60646-1

Round 2

Reviewer 1 Report

  1. Regarding sample size calculations, it is still unclear which variable with a frequency of 50% was used.
  2. After describing that it includes TV viewing and video games, screen time should be used throughout. However, in the discussion, it should be clear whether it is only TV viewing to or total screen time when referring to other studies.
  3. In the methods, include description of anthropometric data collection. In the discussion, the missing data on weights and heights is referred to, but nothing is mentioned about the collection of these data. Finally, anthropometric data allowing to compute BMI was available for how many students?
  4. Also include in data analysis the multiple regression model. Total screen time (per day?) was the dependent variable, but what exactly were the independent variables and in what format? The results of the multiple regression should be detailed in the results, with beta and ‘p’ for each independent variable. The findings also have to be discussed.
  5. The variables in the tables (and in the text) should not be in question format; this is very awkward.
  6. The practice of sports was limited to that with other family members or care givers? Why??? Similarly, breakfast WITH PARENTS only? Unclear.
  7. In Table 3, the full contingency table should be provided.
  8. The results are not discussed in any depth. How can the authors explain that the multiple regression test was not significant, for instance? And how could the advocated surveillance of physical activity change something? If the statement of line 254 refers to data and is not merely impressionistic, then provide the reference.
  9. There are limitations to the study other than missing anthropometric data.
  10. Thorough editing for English would be required. Just to mention a few errors or ambiguous statements:
    1. Line 39: Eating habits, not ‘feeding’
    2. Line 67: ‘Residential homes’ meaning simply home of the family?
    3. Line 104: Delete the two new sentences. What is ‘subjective poverty’? Is it needed here?
    4. Line 236: Unclear sentence; 24% had fruit or juice at breakfast? (Any breakfast frequency?)
    5. Line 277: Knowledge and understanding of what?

Reviewer 2 Report

I approve of the corrections made by the authors to improve the quality of the article.

Author Response

Thank you very much for your help and evaluation of the article.

Reviewer 3 Report

Thank you for the revised version.

Author Response

Thank you very much for your help and evaluation of the article